# Development of a Machine Learning Algorithm to Correlate Lumbar Disc Height on X-rays with Disc Bulging or Herniation

**DOI:** 10.3390/diagnostics14020134

**Published:** 2024-01-06

**Authors:** Pao-Chun Lin, Wei-Shan Chang, Kai-Yuan Hsiao, Hon-Man Liu, Ben-Chang Shia, Ming-Chih Chen, Po-Yu Hsieh, Tseng-Wei Lai, Feng-Huei Lin, Che-Cheng Chang

**Affiliations:** 1Department of Biomedical Engineering, National Taiwan University, Taipei City 10617, Taiwan; killer-ryan-lin@hotmail.com (P.-C.L.); double@ntu.edu.tw (F.-H.L.); 2Department of Neurosurgery, Fu Jen Catholic University Hospital, Fu Jen Catholic University, New Taipei City 24352, Taiwan; 3Graduate Institute of Business Administration, College of Management, Fu Jen Catholic University, New Taipei City 24352, Taiwan; sanchangai@gmail.com (W.-S.C.); st880005@gmail.com (K.-Y.H.); 025674@mail.fju.edu.tw (B.-C.S.); 081438@mail.fju.edu.tw (M.-C.C.); 4Artificial Intelligence Development Center, Fu Jen Catholic University, New Taipei City 24352, Taiwan; 5Department of Radiology, Fu Jen Catholic University Hospital, Fu Jen Catholic University, New Taipei City 24352, Taiwan; 138583@mail.fju.edu.tw; 6Industrial Technology Research Institute (ITRI), Hsinchu City 310401, Taiwan; poyu429@itri.org.tw (P.-Y.H.); lai51613@gmail.com (T.-W.L.); 7Department of Neurology, Fu Jen Catholic University Hospital, Fu Jen Catholic University, New Taipei City 24352, Taiwan; 8PhD Program in Nutrition and Food Science, Fu Jen Catholic University, New Taipei City 24352, Taiwan

**Keywords:** lumbar disc bulging, herniated intervertebral disc, disc height, machine learning, decision tree, plain radiography, magnetic resonance imaging

## Abstract

Lumbar disc bulging or herniation (LDBH) is one of the major causes of spinal stenosis and related nerve compression, and its severity is the major determinant for spine surgery. MRI of the spine is the most important diagnostic tool for evaluating the need for surgical intervention in patients with LDBH. However, MRI utilization is limited by its low accessibility. Spinal X-rays can rapidly provide information on the bony structure of the patient. Our study aimed to identify the factors associated with LDBH, including disc height, and establish a clinical diagnostic tool to support its diagnosis based on lumbar X-ray findings. In this study, a total of 458 patients were used for analysis and 13 clinical and imaging variables were collected. Five machine-learning (ML) methods, including LASSO regression, MARS, decision tree, random forest, and extreme gradient boosting, were applied and integrated to identify important variables for predicting LDBH from lumbar spine X-rays. The results showed L4-5 posterior disc height, age, and L1-2 anterior disc height to be the top predictors, and a decision tree algorithm was constructed to support clinical decision-making. Our study highlights the potential of ML-based decision tools for surgeons and emphasizes the importance of L1-2 disc height in relation to LDBH. Future research will expand on these findings to develop a more comprehensive decision-supporting model.

## 1. Introduction

Lumbar disc bulging or herniation (LDBH) is one of the most common degenerative spinal disorders, leading to nerve compression and radiculopathy [1]. Approximately 10% of patients experiencing low-back pain are diagnosed with LDBH [2]. Large herniated discs can result in severe compression of nerve roots and spinal stenosis, leading to lower-extremity neuralgia, weakness, and numbness and potentially causing various disabilities [3]. To prevent irreversible neurological complications, surgical intervention is needed for patients with severe neurological symptoms [4,5,6,7,8,9,10]. Diagnostic imaging, including X-rays and magnetic resonance imaging (MRI), is often employed to assess the degree of nerve root compression caused by disc herniation and to identify the level of LDBH before surgical intervention [4]. Simple spinal X-rays can offer insights into the parameters of the bony structure; nevertheless, for accurate confirmation of disc herniation and the severity of nerve root compression, spinal MRI is typically necessary [8,11]. MRI offers clear visualization of various spinal structures, including ligaments, facet joints, and discs, making it especially effective for soft-tissue assessment. As a result, MRI is commonly the preferred preoperative evaluation tool for surgeons [12]. However, MRI has limitations, including time consumption, limited accessibility (due to insufficient facilities), and high cost.

Plain radiography (X-ray) is the most commonly used and accessible imaging technique due to its cost-effectiveness and ease of use [13]. Spinal X-rays provide rapid visualization of conditions such as spine fractures, spondylolisthesis, spur formation, and structural deformities. Some degree of soft-tissue degeneration can also be inferred from changes in bony structure [14]. For example, a decreased intervertebral space may suggest degenerative disc changes, and severe spondylolisthesis often coexists with spinal stenosis. Despite advances in imaging technology, the accuracy of X-ray-based diagnostic imaging for LDBH remains questionable [15]. Literature reviews have even noted discrepancies between imaging findings and clinical parameters [16]. Additionally, there is a lack of standardized methods for interpreting the lumbar X-ray images of patients with LDBH. Therefore, the utilization of simple lumbar X-ray imaging to establish an effective method to assist physicians in rapid interpretation is an important yet still poorly understood area.

The emergence of machine learning (ML) has introduced a new perspective for addressing healthcare challenges in medicine and surgical decision-making [17]. Current medical practices incorporate ML methods, which play a crucial role by extracting valuable insights from data without the need for predefined human rules [18,19,20]. ML also aids healthcare professionals in enhancing the quality of care and making precise decisions based on data analysis and interpretation [18]. ML is already extensively employed by physicians and surgeons, encompassing applications in surgical decision support, computer-assisted navigation, and robot-assisted procedures, which have become standard in surgical practice [21,22]. In the current medical landscape, ML algorithms are not only utilized for constructing quantitative classification models but are also widely adopted for medical image interpretation. Various neural network architectures, for instance, have been applied to the interpretation of high-quality CT scans, contributing to image enhancement, restoration, and the generation of 2D/3D medical imagery [23,24,25,26]. These advancements provide healthcare professionals with diverse decision-making references.

Deep learning techniques have initially demonstrated success in the automatic detection and classification of spinal scoliosis. Transfer learning methods, for example, have been proposed to automatically detect and classify spinal scoliosis from spinal X-rays, achieving a level of high accuracy in practical applications [27]. Deep learning techniques have also been applied to identify conditions such as osteopenia and osteoporosis from lumbar X-rays [28]. Natural language processing (NLP) techniques have shown their potential value in spine image analysis. Research has employed the noninvasive identification of curve types in spinal scoliosis from a patient’s 3D back surface, exhibiting a level of high accuracy when compared to expert evaluations from X-ray images [29]. Additionally, NLP techniques have been utilized to identify lumbar spine imaging findings related to low back pain, demonstrating performance comparisons with traditional statistical analysis methods [30]. The diagnosis and treatment of osteoporosis have also benefited from ML techniques. ML models have been employed to predict the T score and identify osteoporotic vertebrae based solely on conventional CT Hounsfield unit (HU) measurements, aiding spine surgeons in accurately assessing osteoporotic spines preoperatively [31]. Furthermore, ML techniques have achieved high levels of accuracy in the classification of conditions such as spondylolisthesis and lumbar lordosis [32].

The use of ML techniques in spine image analysis has increased substantially, showing their potential in enhancing diagnostic accuracy and predictive capabilities across various studies [28,33,34,35]. However, to our knowledge, there is a dearth of research addressing the interpretation or prediction of LDBH outcomes and the relationship between MRI and plain lumbar X-ray. This prospective study aimed to achieve the following objectives: (1) utilize ML techniques to establish the connection between lumbar disc-height measurements from X-ray images and the presence of LDBH detected through MRI scans; and (2) based on this connection, develop a decision-support system capable of predicting LDBH exclusively using X-ray images and basic patient information, including age and sex. This system was designed to develop a decision-support algorithm for clinical practitioners. It aims to identify potential candidates for surgical intervention while minimizing subjective factors and human interference. This process facilitates prompt MRI scheduling for individuals in need of surgical treatment.

## 2. Materials and Methods

### 2.1. Participants and Study Design

A total of 662 patients who underwent lumbar spine MRI at Fu Jen Catholic University Hospital, Taipei, Taiwan, between January 2020 and December 2020 were retrospectively analyzed. Patients were included if they had undergone both lumbar spine MRI studies and X-rays, with the time interval between the two not exceeding 3 months. Patients were excluded if they lacked lumbar spine X-rays or had undergone previous lumbar fixation or fusion surgery. Patients with pathological factors, including vertebral fracture and spondylodiscitis, were also excluded from this study. All imaging was performed using the same equipment and imaging site. A total of 662 patients who underwent both lumbar spine MRI and X-ray were eventually included in this study. Sixty-eight patients with incomplete studies, 37 with previous spine surgery, 21 with vertebral body fractures, 3 patients with extremely blurred X-ray, and 8 patients with spondylodiscitis or other congenital spine diseases were all excluded. Moreover, X-rays from 67 patients were also excluded due to potential interference, such as severe scoliosis, excessive obesity, and issues with imaging quality. These interferences resulted in errors in the image segmentation produced by the measurement software, leading to the exclusion of certain measurement data that exceeded the upper limits of normal anatomical structures. The L1-2, L2-3, L3-4, L4-5, and L5-S1 anterior and posterior disc heights of the included patients were measured from the lateral view of their spinal X-rays, and the measurements were rechecked by one experienced neurosurgeon and one neurologist. Finally, a total of 458 patients were used for analysis (Figure 1). In total, 13 clinical and imaging variables were collected. The protocol of this study was evaluated and deemed acceptable by the Research Ethics Review Committee of the Fu Jen Catholic University Hospital (No. FJUH110121).

### 2.2. Definition of Disc Bulging, Protrusion, and Herniated Disc

Disc bulging and protrusion are defined as the presence of fibrous tissue on the dorsal side of the disc annulus that extends beyond the posterior edge of the vertebral endplates, leading to a reduction in the volume of the spinal canal or the occupation of the foramen space. In addition, the migration of disc material, including the nucleus pulposus, endplate cartilage, and annulus fibrosus, can also cause a reduction in the neural canal space. In clinical practice, we commonly use MRI to diagnose intervertebral disc protrusion and to assess whether there is any compression of the nerves. When a patient’s lumbar spine MRI showed disc protrusion and compression on the spinal canal in any segment from L1-S1, the patient was classified into the LDBH group.

### 2.3. Definition and Measurement of Disc Height

In this study, disc height was defined as the distance between the corner point of the vertebral body and the point of its orthogonal projection on the endplate of the adjacent vertebral body (Figure 2). For example, in the figure, the distance between corner point C and its orthogonal projection point E is defined as the anterior disc height. In the same way, the length from corner point B to its projection point F is defined as the posterior disc height. The anterior height and posterior height were measured separately at the L1-2, L2-3, L3-4, L4-5, and L5-S1 levels. Consequently, 10 measurement data points were collected from every included patient. In most studies, disc height is preferentially measured as the length between adjacent corner points (bd or ac), but in severe spondylolisthesis patients, the length may be overestimated [36,37,38,39,40,41].

#### Measurement of Disc Height with BiLuNet

BiLuNet is a novel multipath convolutional neural network designed for semantic segmentation in X-ray images, and it has been employed in medical applications such as lumbar vertebrae, sacrum, and femoral head segmentation. One of the significant benefits of BiLuNet is its capability to produce a high level of accuracy in segmenting and shape fitting lumbar vertebrae, sacrum, and femoral heads on X-ray images [42]. This study employs BiLuNet for the localization of intervertebral discs in X-ray images and measures their heights. The measured values are then applied in the subsequent machine learning workflow (Figure 3).

### 2.4. Statistical Analysis

This study used five ML algorithms, namely, least absolute shrinkage and selection operator (LASSO), multivariate adaptive regression splines (MARS), decision tree, random forest, and extreme gradient boosting (XGBoost), to construct predictive models for classifying LDBH patients and to evaluate the importance of different measurements of disc height.

#### 2.4.1. LASSO

LASSO is one of the best regression methods for both variable selection and regularization for addressing the overfitting problem and obtaining accurate results. It uses a penalty parameter to shrink small coefficients toward zero during model estimation [43,44].

#### 2.4.2. MARS

The MARS technique, standing for multivariate adaptive regression splines, represents a sophisticated and nonlinear approach to spline regression, a variant of regression analysis. This method distinctively employs a multitude of piecewise linear segments, commonly known as splines, each characterized by varying slopes or gradients. In its operation, MARS treats every data sample as a ‘knot’, segmenting the dataset into multiple parts. This partitioning facilitates the execution of linear regression analysis in a stepwise manner, focusing on each divided section individually. The process of knot determination in MARS is twofold: initially, a forward selection procedure is employed. This step involves the comprehensive screening of all potential basic functions along with their respective knots. Following this, a backward elimination strategy is implemented, where these basic functions are systematically removed. The aim of this backward phase is to refine and optimize the combinations of the remaining knots, ensuring that the most effective and representative model is achieved [45].

#### 2.4.3. Decision Tree

A decision tree is a supervised learning method in which a tree-like structure is used to draw conclusions about some observations. Numerous different trees can be developed: regressive binary partition trees use an algorithm that can perform classification for regression problems; models where the target variables are continuous data are called regression trees; and models where the target variables are discrete data are known as classification trees [46]. The latter form was used in this study.

#### 2.4.4. Random Forest

Random forest is an ML model in which the integration of multiple decision trees can improve the high variability in the original decision tree model. A random forest is constructed by first generating multiple decision trees [47]. Then, the final prediction is obtained by voting for the result of the resulting decision tree.

#### 2.4.5. Extreme Gradient Boosting

XGBoost is a popular algorithm for both regression and classification tasks. It improves the integration of the gradient boosting algorithm to obtain better performance in ML tasks. The XGBoost algorithm uses parallel, distributed learning via fast, well-optimized, and scalable algorithms [48]. Ensemble algorithms can enhance the model’s performance through the addition of new models until the performance of the model no longer advances.

#### 2.4.6. ML Workflow and Implementation Details

In this study, all methods were conducted in R software version 4.0.3 and RStudio version 1.4.1103. The algorithms for the methods are based on the relevant R packages. LASSO was implemented using the “glmnet” package, version 4.1-1. MARS was implemented using the “earth” package, version 5.3.2. The decision tree was implemented using the “rpart” package, version 4.1-15. Random forest was implemented with the “randomForest” package, version 4.6-14. XGBoost was implemented with the “XGBoost” package, version 1.5.0.1. The “caret” package, version 6.0-90, was used to evaluate the importance of different factors in each method.

This research introduced a structured machine learning workflow (Figure 3), primarily aimed at evaluating and ranking the importance of features. Initially, the raw data underwent preprocessing to ensure its quality and integrity. After this preprocessing phase, the dataset was partitioned into training and testing sets, with the training set accounting for 80% of the total data and the testing set accounting for the remaining 20%. Subsequently, several classification models were employed, and each model was subjected to 10-fold cross-validation to assess its performance. The 10-fold cross-validation method was adopted to determine the optimal hyperparameters for each model, as this approach provides a more consistent evaluation of various methods. This study assessed the performance of these machine learning algorithms using metrics such as accuracy, recall, specificity, precision, and F1 score.

To estimate the best parameter set for the development of the five models, the “caret” package in R was utilized to tune the relevant hyperparameters. The initial models were constructed using default settings. Afterward, feature importance was extracted independently from each model. An average ranking method was then applied, considering the outputs from all models, to offer a comprehensive and objective assessment of feature significance. This study employed the “varImp” function from the caret package to ascertain the relative significance of predictors in each model. Through this function, we sorted the predictors based on their relative contribution to the importance of variables for every model.

## 3. Results

The 12 variables considered as impact factors for LDBH (Y) in patients are shown in Table 1. The average age of patients who had LDBH was 60.00 ± 14.00 years, while the average age of the non-LDBH patients was 58.98±14.14 years. In the LDBH group, there were 133 males (51.4%) and 126 females (48.6%). In the non-LDBH group, 100 patients (50.3%) were male, and 99 patients (49.7%) were female. The anterior and posterior mean disc heights from L1/2 to L5/S1 were measured and are listed in Table 1.

This study used LASSO, MARS, decision tree, random forest, and XGBoost to construct predictive models to evaluate the measured parameters. Table 2 shows the model performance of the five methods with the validation dataset and testing dataset. The average F1 score values of the LASSO, MARS, decision tree, random forest, and XGBoost models were 0.706, 0.778, 0.569, 0.729, and 0.706 with the testing dataset, respectively. The MARS model provided the highest average F1 score value, followed by the random forest, LASSO, XGBoost, and decision tree models.

The importance ranking of each variable generated by the LASSO, MARS, decision tree, random forest, and XGBoost methods is shown in Figure 4. In this figure, it can be seen that each model provides a different sequence for the relative importance ranking of each variable. For example, the posterior L3-4 disc height is the most important variable in the model constructed with LASSO regression, and the second most important variable is anterior L4-5 disc height. Computing the average rank to explore the importance of the variables shows that the three most relatively important variables are the anterior L4-5 disc height, anterior L1-2 disc height, and posterior L1-2 disc height, which can provide certain insights into their role in LDBH.

## 4. Discussion

To our knowledge, few studies have attempted to use ML methods to predict lumbar disc diseases from spinal X-rays. This study revealed that the L4-5 anterior disc height and L1-2 anterior and posterior disc heights were the top three parameters that helped us to best predict LDBH using plain lumbar X-ray imaging. The classification and regression tree (CART) method generated the best and most promising classification results and provided an output of six clinical features that were critical for the prediction of LDBH. Decision rules for the prediction of LDBH according to the plain X-ray findings were also constructed, as shown in Figure 5.

Disc bulging, protrusion, and even herniated intervertebral discs (HIVDs) are the most common spinal degenerative diseases and can only be confirmed by spinal MRI. Disc degeneration is the beginning stage of LDBH and is strongly associated with facet degeneration, foramen, and lateral recess narrowing, spondylolisthesis, and spinal stenosis. As the degeneration worsens, the symptoms can change from low-back pain to severe neuralgia, claudication, and even cauda equina syndrome. Access to facilities and the costs of MRI limit early detection. This delay may result in irreversible neurological deficits. Decreased disc height, spur formation, spondylolisthesis, and an abnormal range of motion between adjacent vertebral bodies are features of disc degeneration on spinal X-ray, but it is difficult to diagnose LDBH and its related neural structure compression. In this situation, ML can help clinicians construct a decision tree model to predict LDBH simply with plain lumbar X-ray findings and gain new insights for future studies. In areas without access to MRI, such as remote or primary care clinics, this decision-support system may assist high-risk patients in receiving timely examination and treatment.

The strength of artificial intelligence (AI) has grown in the field of neurosurgery. ML can help clinicians via automated computer systems that predict outputs through mathematics [49]. Currently, ML applications in spinal surgery include image classification (e.g., the detection of compression fractures on CT or MRI [50,51,52], the construction of risk models, and decision support tools) and diagnostic assistance [53,54,55,56,57]. Trinh et al. [58] reported that several deep learning methods can be used to develop a diagnostic algorithm for automatically recognizing spondylolisthesis based on lumbar X-ray images. In one retrospective cohort study, a deep learning method was applied to predict adolescent idiopathic scoliosis (AIS) with standing posteroanterior X-rays [59].

However, few studies have focused on using plain X-ray findings to identify potential patients who need further MRI studies or surgical intervention due to LDBH. The relationship between disc degenerative diseases and disc height remains controversial. One in vitro study with partial discectomy of 15 human lumbar discs demonstrated that the change in disc height was associated with the mass of central disc tissue, and disc height decreases were also related to radial disc bulge. Another study showed that the influences of disc level and degree of degeneration on mechanical behavior are not significant [60,61]. However, in another retrospective study, researchers investigated the relationship between disc morphology and bulging by using MRI scans. They revealed that disc height/depth was significantly associated with the outcome of disc herniation, especially at the L3-4 and L4-5 levels [37]. Our study, which differed from previous methods, attempted to identify possible significant measurements of different disc heights on spine X-rays using ML methods. The age, sex, BMI, and anterior and posterior disc heights at L1-2, L2-3, L3-4, L4-5, and L5-S1 were measured and analyzed using ML methods. The performance of the ML methods, listed in Table 2, demonstrates that these methods are not inferior to traditional LASSO regression.

The importance of the anterior and posterior disc height at different levels was ranked by the five ML methods and is listed in Figure 4. The parameter with the highest average rank was the anterior disc height of L4-5. This finding is compatible with previous studies and clinical MRI results [61]. Disc degeneration can occur at any lumbar spine level but is more commonly found at L4-5 [61]. In the context of this study, age, gender, and BMI are relatively ranked in the middle to lower range in terms of average importance for predicting overall LDBH risks. This suggests that the predictive significance of age, sex, and BMI for LDBH may not be as substantial as is commonly perceived. When only considering patient X-ray information, the measurement of disc height for each segment appears to hold more reference value in predicting LDBH compared to the actual measurements of age, sex, and BMI.

Disc degeneration mainly occurs at L3-4, 4-5, and L5-S1 and frequently results in LDBH. Clinical physicians can easily and directly diagnose disc degeneration through degenerative findings at these three levels on spinal X-rays. However, the average ranking from the ML methods revealed that the height at L1-2 was a potentially more predictive factor than the heights at L3-4 and L5-S1. ML methods may uncover the importance of disc height at L1-2 to predict overall LDBH risks. ML methods may uncover the importance of disc height at L1-2 to predict overall LDBH risks. This unconventional finding seems to have clinical relevance as well. Decreased disc height is always present from the beginning of disc degeneration, and the L1-2 disc degeneration rate is the lowest among all levels in the lumbar discs. More severe degeneration in L1-2 hints at a higher risk of other levels of disc bulging or herniation. The thoracolumbar junction, particularly the segment from T12 to L2, is considered a transition zone between the relatively less mobile thoracic vertebrae and the more mobile lumbar vertebrae. Segments within this region bear significant stress, contributing to over 60% of compressive fractures occurring in the vertebral bodies of T12-L2 [62]. The reduced disc height at L1-2 may also suggest that patients’ spines experience greater stress, leading to an accelerated rate of degeneration in the lumbar region compared to normal individuals. This could explain the close relationship between L1-2 disc height and the overall lumbar disc body height.

In the past, plain X-ray could only be used to identify simple bone structure problems, such as spondylosis, spondylolisthesis or compression fractures. The advantages of X-ray over other imaging modalities are that it is less expensive, and its images can be assessed and judged more easily and faster. Our study results, although preliminary, show the development of a novel decision-support system that enables clinical physicians in remote or primary care clinics lacking access to MRI to use simple and fast spinal X-ray screening to identify high-risk patients and promptly refer them to hospitals with MRI capabilities. The purpose and results of this study do not replace the critical role of MRI in diagnosing lumbar degenerative diseases. However, from the perspective of diagnostic assistance and decision support, it can be seen as a valuable contribution.

### 4.1. Clinical Implications

This pilot study has some clinical implications. First, our study aimed to provide a tool for identifying potential parameters to rapidly identify the possible risk of LDBH based on X-ray findings. By analyzing the results, we provided a potential method for physicians to quickly refer patients in a timely manner. Although previous studies have provided controversial conclusions about the correlation between the parameters found in spinal X-rays and the definite diagnosis found in MRI, this study still used multiple analysis methods, including ML methods, to clarify the relationship between X-ray findings and MRI diagnosis. If a reliable decision tree can be made in the future, X-rays can identify high-risk patients in rural areas and shorten the MRI waiting list in medical centers. Although the patient number is limited, the result is positive and hints that X-rays can potentially provide more information. Second, this study used ML methods to assess LDBH and revealed a new perspective. Using this method could improve the diagnosis of LDBH and allow hospitals with insufficient equipment or long MRI schedules to select potential high-risk patients. In future works, the model can be combined with other clinical parameters, including occupation, and any other information from plain X-rays. This pilot study recommends the model as a potential primary benchmarking tool for use in the screening of LDBH in outpatient clinics.

### 4.2. Limitations and Work in Progress

This study presents several limitations. Primarily, the sample size we utilized is not very large and is sourced exclusively from a single institution. In addition to the data processing discussed, our research employed a range of statistical techniques pertinent to ML. As outlined in Section 2.4, our approach integrated methods such as LASSO, ridge regression, decision trees, random forests, and XGBoost. We further incorporated a 10-fold cross-validation approach to ensure a more stable evaluation of our models. The amalgamation of these methodologies reinforces the robustness and credibility of our findings. Second, our analysis focused exclusively on disc height, sex, BMI, and age. To broaden the scope of future investigations, it is imperative to explore a more extensive array of parameters that can be derived from X-rays. Consequently, the generalizability of our findings should be interpreted with caution, and further studies are warranted. In recognition of these constraints, we are actively pursuing several enhancements to address these limitations. These include the following: (1) Expanding the sample size to bolster the reliability and comprehensiveness of our results. (2) Collaborating with multiple institutions to access a more diverse and representative dataset. (3) Integrating additional parameters for a more in-depth analysis, including the clinical sign and symptoms of each induvial to enhance the accuracy of predicting MRI outcomes. Our ongoing efforts are dedicated to fortifying the robustness and applicability of our study in the pursuit of more extensive and generalizable insights.

## 5. Conclusions

Our study utilized ML-based methods to correlate lumbar disc height on X-rays with LDBH and attempted to construct a potential clinical decision-making tool to support the diagnosis of LDBH based on X-ray imaging parameters. The results revealed that the anterior L4-5 and anterior L1-2 disc heights, as well as posterior L1-2 disc heights, were the three most important variables in diagnosing potential LDBH. The importance of the L1-2 disc height was also revealed for the first time in this study. While still only preliminary, the current study attempts to correlate lumbar disc height on X-rays with LDBH and construct a potential algorithm for screening high-risk LDBH patients. Our results represent an exploratory study of LDBH risk using MRI as the gold standard, and further studies will include more patients and analyze more parameters to construct a more reliable decision-supporting model.

## Figures and Tables

**Figure 1 diagnostics-14-00134-f001:**
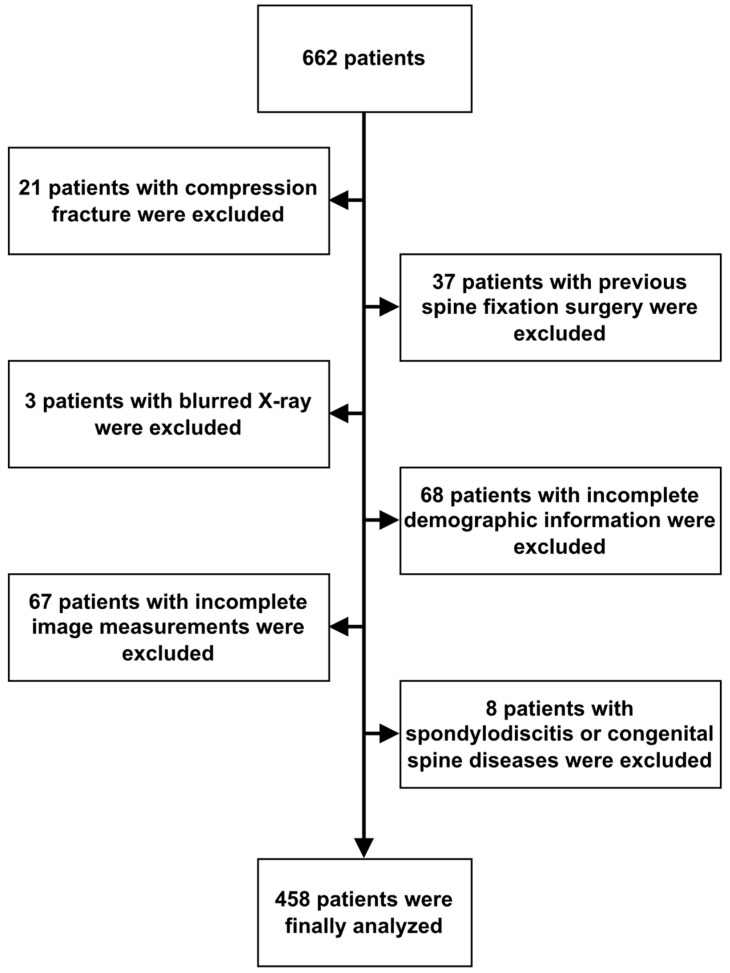
Algorithm of case identification.

**Figure 2 diagnostics-14-00134-f002:**
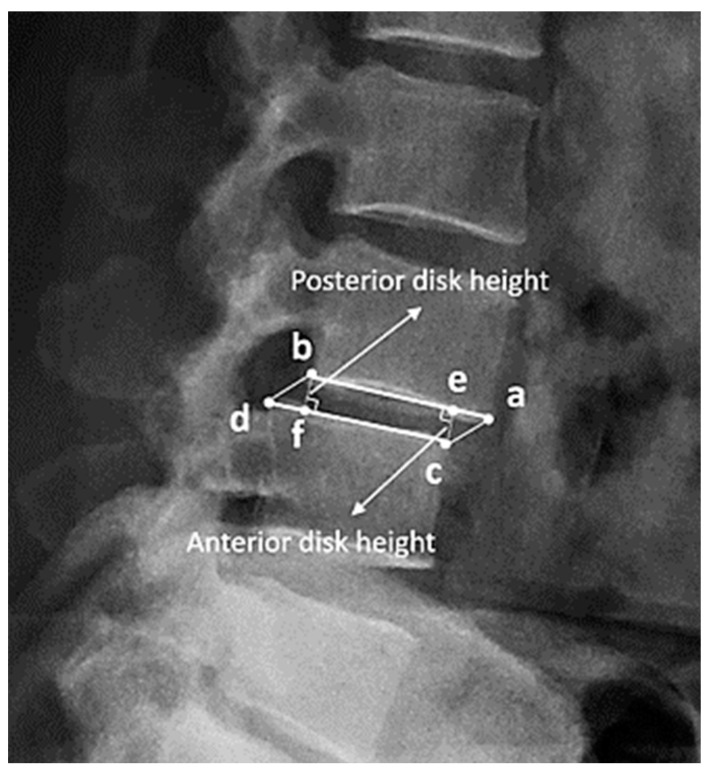
(a–d) are the corner points of the adjacent vertebral body. (e) is the orthogonal projection point on the endplate (a and b) of (c), and (f) is the orthogonal projection point of (b). The disc height is defined as the distance between the corner point of the vertebral body and the point of its orthogonal projection on the endplate of the adjacent vertebral body. For example, the anterior disc height is the length from (e) to (c), and the posterior disc height is the length from (b) to (f).

**Figure 3 diagnostics-14-00134-f003:**
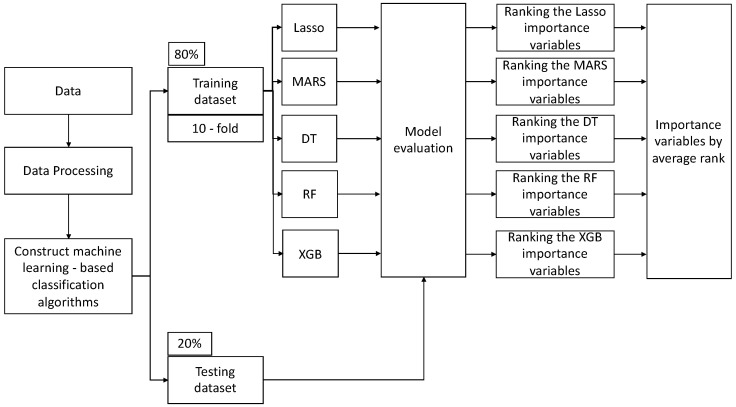
Machine learning (ML) analytical workflow in our study.

**Figure 4 diagnostics-14-00134-f004:**
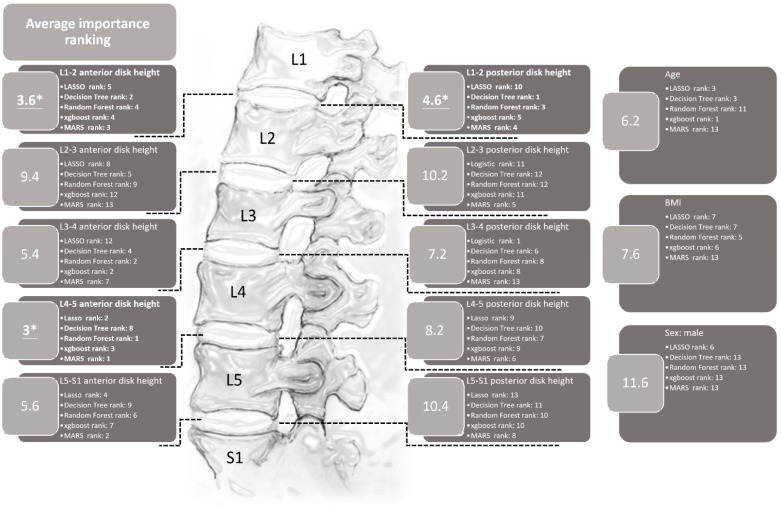
The importance rankings of each variable using the LASSO, ridge, decision tree, random forest, XGBoost, and MARS methods. This figure reveals that the anterior L4-5 and the anterior and posterior L1-2 disc heights are the three most important variables in terms of average ranking. * The top three most important variables.

**Figure 5 diagnostics-14-00134-f005:**
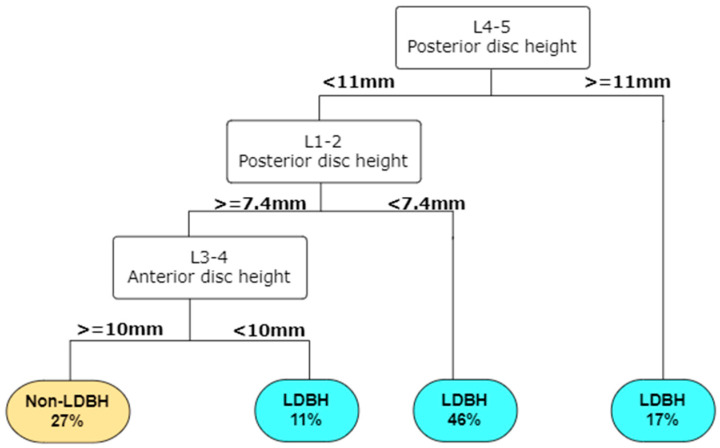
A decision tree can be built in RStudio according to the importance of the input parameters. For example, according to this decision tree, a patient has a 46% risk of LDBH if his posterior L4-5 disc height is less than 11 mm and his L1-2 posterior disc height is less than 7.4 mm. This decision tree helps identify the risk of lumbar disc bulging, or herniation simply from information derived from lumbar spine X-ray. Abbreviations: LDBH: Lumbar disc bulging or herniation.

**Table 1 diagnostics-14-00134-t001:** Subject demographics and clinical characteristics.

	LDBH *n* = 259	Non-LDBH *n* = 199
Age (mean ± SD)	60.00 ± 14.00	58.98 ± 14.14
Sex = Male (%)	133 (51.4)	100 (50.3)
BMI (mean ± SD)	25.76 ± 4.09	26.33 ± 4.21
Disc height measurement (mean ± SD) (mm)		
Disc height L1-2 anterior	9.69 ± 2.12	9.36 ± 2.19
Disc height L1-2 posterior	7.46 ± 1.60	7.33 ± 1.50
Disc height L2-3 anterior	10.74 ± 2.40	10.26 ± 2.16
Disc height L2-3 posterior	8.04 ± 2.04	7.68 ± 1.79
Disc height L3-4 anterior	11.83 ± 2.68	11.44 ± 2.78
Disc height L3-4 posterior	8.96 ± 2.74	8.28 ± 2.12
Disc height L4-5 anterior	12.88 ± 8.90	11.47 ± 3.76
Disc height L4-5 posterior	10.99 ± 9.53	9.68 ± 5.99
Disc height L5-S1 anterior	15.10 ± 7.27	14.57 ± 7.92
Disc height L5-S1 posterior	9.59 ± 9.31	8.51 ± 5.28

Abbreviations: LDBH: lumbar disc bulging or herniation.

**Table 2 diagnostics-14-00134-t002:** Performance of the LASSO, MARS, decision tree, random forest, and XGBoost methods.

Method	Avg_Accuracy	Avg_Recall	Avg_Precision	Avg_Specificity	Avg_F1
Testing Dataset					
LASSO Regression	0.615	0.857	0.600	0.333	0.706
MARS	0.689	0.924	0.676	0.357	0.778
Decision Tree	0.516	0.592	0.547	0.429	0.569
Random Forest	0.655	0.794	0.675	0.458	0.729
XGBoost	0.615	0.857	0.600	0.333	0.706

Abbreviations: avg: average.

## Data Availability

Data are available upon reasonable request from the authors with the permission of the local ethics committee.

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
