# Peer review of "Development of a Machine Learning Algorithm to Correlate Lumbar Disc Height on X-rays with Disc Bulging or Herniation"

_diagnostics, 2024, doi:10.3390/diagnostics14020134_

Round 1

Reviewer 1 Report (Previous Reviewer 2)

Comments and Suggestions for Authors

We appreciate the authors’ efforts in revising the manuscript, and they did good work; please see below for minor concerns.

1 Could authors show some sample raw images with corresponding GT labels of Fu Jen Catholic University Hospital, Taipei, Taiwan, used for experiments in section “2. Materials and Methods”?

2 Please give the source reference of Figure 4.

Author Response

Dear reviewer:

Thank you for the nice and detail review for concerning our manuscript. Please accept our sincerely appreciation to the editor for his/her positive and encouraging comments on the manuscript. We have made a point-to-point revision and response to all your reviewer’s comments. The reasons and revisions are listed in the following tables to each reviewer, respectively. In the revise manuscript, all the changes are highlighted in Red. Thanks for the advices, we have incorporated the suggestions into the revised manuscript. Also, this study is much improved as a result. Replies to specific comments and described in detail as follow (in blue color).

Sincerely,

Che-Cheng Chang

Reviewer 2 Report (New Reviewer)

Comments and Suggestions for Authors

This is a very interesting and worthwhile study. The clinical impact of these machine learning techniques is notable. Thank you for your research. 

Overall comments-

The inclusion/exclusion criteria need to be clarified and the rationale explained briefly. 

Good explanations of the five machine learning outcomes and the statistical methods. 

Overall, very well written but I would review the level of repetition in places. Please see document attached for more specific comments. 

Comments on the Quality of English Language

High quality of writing, some suggestions for improvement attached in document

Author Response

Dear reviewer:

Thank you for the nice and detail review for concerning our manuscript. Please accept our sincerely appreciation to the editor for his/her positive and encouraging comments on the manuscript. We have made a point-to-point revision and response to all your reviewer’s comments. The reasons and revisions are listed in the following tables to each reviewer, respectively. In the revise manuscript, all the changes are highlighted in Red. Thanks for the advices, we have incorporated the suggestions into the revised manuscript. Also, this study is much improved as a result. Replies to specific comments and described in detail as follow (in blue color).

Sincerely,

Che-Cheng Chang

Reviewer 3 Report (New Reviewer)

Comments and Suggestions for Authors

Dear, Editor of Diagnostics

            Thank you for give me an opportunity to review this article name “Development of a Machine Learning Algorithm to Correlate Lumbar Disc Height on X-rays with Disc Bulging or Herniation”.  My comment and suggestion as the following:

1.    - In introduction line number 58-60: I do not agree with sentence “time consumption, limited accessibility and high cost”.  Because, now a day, even in my country MRI is available and not to expensive.  In this study, only one part of MRI on Lumbosacral was used for this study, therefore it should not expensive and no time consuming.

2.    - In introduction line number 114 to 116: I do not agree with this sentence “This system was designed to assist clinical practitioners in regions with limited access to MRI facilities or prolong MRI waiting list by identifying potential candidates”.  Normally, clinician will make decision about the scheduling of MRI based on the severity of clinical of patient e.g. cauda equina syndrome should be emergency condition and need emergency MRI. Other non-emergency condition, no need for change the schedule of MRI.  Furthermore, this study did not correlate with clinical symptoms of patients.

3.    - In material and methods on 2.1 Participants and study design: Author should give more detail about clinical symptoms of patients which included in this study.  Because some patients has disc bulging, protrusion, or herniation have no clinical symptoms.

4.   - In Material and methods on 2.2 definition of disc bulging, protrusion, and herniated disc line number 141-143:  Author should give more detail about the part of MRI using for diagnosis of disc bulging, protrusion, and herniation such as sagittal cut or axial cut.

5.     -In table 1: Author should give detail about spinal level of disc bulging, disc protrusion, or herniation in LDBH patients.  In addition, clinical symptoms of LDBH patients such as spinal stenosis or radiculopathy or health related parameter e.g. pain score.  Because if patient do not have symptom, it will be incidental finding during MRI.

Comments on the Quality of English Language

Quality of English is very good

Author Response

Dear reviewer:

Thank you for the nice and detail review for concerning our manuscript. Please accept our sincerely appreciation to the editor for his/her positive and encouraging comments on the manuscript. We have made a point-to-point revision and response to all your reviewer’s comments. The reasons and revisions are listed in the following tables to each reviewer, respectively. In the revise manuscript, all the changes are highlighted in Red. Thanks for the advices, we have incorporated the suggestions into the revised manuscript. Also, this study is much improved as a result. Replies to specific comments and described in detail as follow (in blue color).

Sincerely,

Che-Cheng Chang

Round 2

Reviewer 3 Report (New Reviewer)

Comments and Suggestions for Authors

Dear, authors

   Thank you for your response to my comment.  I have no further comments.

Comments on the Quality of English Language

Quality of English is very good.

This manuscript is a resubmission of an earlier submission. The following is a list of the peer review reports and author responses from that submission.

Round 1

Reviewer 1 Report

Comments and Suggestions for Authors

The manuscript “Lateral-PLIF for spinal arthrodesis: concept, technique, results, complications, and outcomes” by  Pao-Chun Lin et al. to aimed to use ML methods to identify the factors associated with LDBH and disc height reduction, and attempt to establish a clinical diagnostic decision tree tools for supportive diagnosis of LDBH based on lumbar X-ray findings.

The authors used five machine learning methods, namely, lasso regression, ridge regression, decision tree, random forest and eXtreme Gradient Boosting, to analyze the data of 69 patients who underwent both lumbar spine MRI and X-ray. They found that L4-5 posterior disk height, age, and L1-2 anterior disk height were the top predictors of LDBH, and they developed a decision tree algorithm to support clinical decision-making.

Below are my comments and remarks regarding the manuscript:

1. The article has a small sample size of 69 patients from a single institution, which limits the generalizability and validity of the results. 

2. Small non-HIVD group: This is an observation that points out a potential limitation of the study, which is the small sample size of the non-HIVD group (n=12) compared to the HIVD group (n=57), which may affect the validity and generalizability of the results.

3. The article also lacks a validation or test set to evaluate the accuracy and robustness of the models.

4. The article only considers disk height, age and sex as the input variables for the machine learning models, which may not capture the complexity and heterogeneity of LDBH. 

5. The article does not include other potential factors such as disk morphology, spinal alignment, facet joint degeneration, ligamentum flavum hypertrophy, etc.

6. The article does not provide any comparison or correlation between the X-ray findings and the MRI findings, which are the gold standard for diagnosing LDBH. The article also does not report any clinical outcomes or symptoms of the patients with LDBH.

7. The authors did not provide a explanation of what HIVD in the text 

8. The idiopathic and degenerative scoliosis should be included in the exclusion criteria

9. How the X-ray images were obtained, such as whether they were taken in a standing, sitting, or lying position, and whether they were taken in a neutral, flexed, or extended posture. 

Author Response

(The authors gave the same response as above.)

Reviewer 2 Report

Comments and Suggestions for Authors

In this paper, authors aimed to identify the factors associated with LDBH, including the disc height, and try to establish a clinical diagnostic tool for supportive diagnosis of LDBH based on lumbar X-ray findings. In this study, five machine-learning methods, including lasso regression, ridge regression, decision tree, random forest and eXtreme Gradient Boosting was application and integration to identify important variables in predicting LDBH from lumbar spine X-rays. The authors did good work and were interested in the readers. The following review comments are recommended, and the authors are invited to explain and modify.

1 Novelty is confusing. A highlight is required. The main contributions of the manuscript are not clear. The main contributions of the ‎article must be very clear and would be better if summarize ‎them into 3-4 points at the ‎end of the introduction.‎

2 At the end of the second section, the authors should discuss the limitations of the current related work. Then, they should discuss how they overcame these limitations in their proposed system.

3 The introduction section needs to be improved. An introduction is an important road map for the rest of the paper that should be consist of an opening hook to catch the researcher's attention, relevant background study, and a concrete statement that presents main argument but your introduction lacks these fundamentals, especially relevant background studies. This related work is just listed out without comparing the relationship between this paper's model and them; only the method flow is introduced at the end; and the principle of the method is not explained. To make soundness of your study must include these latest related works. Authors also need to justify the importance of their article and cite all of them to make a critical discussion that makes a difference from others' work.

I (2022). An Effective WSSENet-Based Similarity Retrieval Method of Large Lung CT Image Databases. KSII Transactions on Internet & Information Systems, 16(7). doi: 10.3837/tiis.2022.07.013

II (2022). The algorithm of stereo vision and shape from shading based on endoscope imaging. Biomedical Signal Processing and Control, 76. doi: 10.1016/j.bspc.2022.103658

III (2022). Improved Feature Point Pair Purification Algorithm Based on SIFT During Endoscope Image Stitching. Frontiers in Neurorobotics. doi: 10.3389/fnbot.2022.840594

IV (2022). 2D/3D Multimode Medical Image Registration Based on Normalized Cross-Correlation. Applied Sciences, 12(6). doi: 10.3390/app12062828

4 Methodology should provide a flowchart of the whole work. ‎This will help the reader to get a better understanding of what is going on in the proposed ‎system.‎

5 There are the latest deep learning techniques for high-level feature extraction to give high accuracy, but why did the authors use basic machine-learning methods?

6 Sciatica frequently results from a herniated disc in the lower back especially L4-5 and L5-S; then what is the findings of authors based on this ML research?

7 Authors should define all evaluation metrics.

8 Authors should mention the implementation challenges.

9 Moreover, it should be noticed that the clinical appliance has to be decided by medicals since the existing differences between the real image and the one generated by the proposed model could be substantial in the medical field.

Comments on the Quality of English Language

 Minor editing of English language required.

Author Response

(The authors gave the same response as above.)

Reviewer 3 Report

Comments and Suggestions for Authors

The authors propose the use of hand-designed features from X-ray images of the spine and machine learning in order to solve the binary classification problem of Lumbar disk bulging or herniation (LDBH).

The authors suggest that the motivation for this research is that the use of the golden standard of MRI may be unavailable, expensive or time-consuming. However, this is not sufficiently justified as patients suffering with spinal issues would prefer the use of a modern tomographic modality which provides the facilities of 3D visualization and automated measurements, so as to perform accurate diagnosis; indeed, most city medical facilities would have access to MRI scanners, with the associated procedure being completed in 30-40 minutes. In this respect, the research is not sufficiently justified.

The manuscript is lacking a substantial state of the art (SOA) review in regards to the use of machine learning in the broad field of spine image analysis. A quick review reveals examples of potential sources for SOA analysis as follows:

D’antoni F., Russo F., Ambrosio L., Bacco L., Vollero L., Vadalà G., Merone M., Papalia R., Denaro V. Artificial Intelligence and Computer Aided Diagnosis in Chronic Low Back Pain: A Systematic Review (2022) International Journal of Environmental Research and Public Health, 19 (10), art. no. 5971,  DOI: 10.3390/ijerph19105971

Jujjavarapu C., Pejaver V., Cohen T.A., Mooney S.D., Heagerty P.J., Jarvik J.G. A Comparison of Natural Language Processing Methods for the Classification of Lumbar Spine Imaging Findings Related to Lower Back Pain (2022) Academic Radiology, 29, pp. S188 - S200, DOI: 10.1016/j.acra.2021.09.005

Samadi B., Raison M., Mahaudens P., Detrembleur C., Achiche S. A preliminary study in classification of the severity of spine deformation in adolescents with lumbar/thoracolumbar idiopathic scoliosis using machine learning algorithms based on lumbosacral joint efforts during gait (2022) Computer Methods in Biomechanics and Biomedical Engineering DOI: 10.1080/10255842.2022.2117547

Amin A., Abbas M., Salam A.A. Automatic Detection and classification of Scoliosis from Spine X-rays using Transfer Learning (2022) 2022 2nd International Conference on Digital Futures and Transformative Technologies, ICoDT2 2022,  DOI: 10.1109/ICoDT255437.2022.9787480

Masood R.F., Taj I.A., Khan M.B., Qureshi M.A., Hassan T. Deep Learning based Vertebral Body Segmentation with Extraction of Spinal Measurements and Disorder Disease Classification (2022) Biomedical Signal Processing and Control, 71, art. no. 103230 DOI: 10.1016/j.bspc.2021.103230

Nam K.H., Seo I., Kim D.H., Lee J.I., Choi B.K., Han I.H. Machine learning model to predict osteoporotic spine with hounsfield units on lumbar computed tomography (2019) Journal of Korean Neurosurgical Society, 62 (4), pp. 442 - 449 DOI: 10.3340/jkns.2018.0178

Adankon M.M., Dansereau J., Labelle H., Cheriet F. Non invasive classification system of scoliosis curve types using least-squares support vector machines (2012) Artificial Intelligence in Medicine, 56 (2), pp. 99 - 107 DOI: 10.1016/j.artmed.2012.07.002

Koompairojn S., Hua K., Hua K.A., Srisomboon J. Computer-aided diagnosis of lumbar stenosis conditions (2010) Progress in Biomedical Optics and Imaging - Proceedings of SPIE, 7624, art. no. 76241C DOI: 10.1117/12.844545

D’antoni F., Russo F., Ambrosio L., Bacco L., Vollero L., Vadalà G., Merone M., Papalia R., Denaro V. Artificial Intelligence and Computer Aided Diagnosis in Chronic Low Back Pain: A Systematic Review (2022) International Journal of Environmental Research and Public Health, 19 (10), art. no. 5971 DOI: 10.3390/ijerph19105971

Bin Zhang, Keyan Yu, Zhenyuan Ning, Ke Wang, Yuhao Dong, Xian Liu, Shuxue Liu, Jian Wang, Cuiling Zhu, Qinqin Yu, Yuwen Duan, Siying Lv, Xintao Zhang, Yanjun Chen, Xiaojia Wang, Jie Shen, Jia Peng, Qiuying Chen, Yu Zhang, Xiaodong Zhang, Shuixing Zhang, Deep learning of lumbar spine X-ray for osteopenia and osteoporosis screening: A multicenter retrospective cohort study, Bone, Volume 140, 2020, 115561, ISSN 8756-3282, https://doi.org/10.1016/j.bone.2020.115561.

W. Katherine Tan, Saeed Hassanpour, Patrick J. Heagerty, Sean D. Rundell, Pradeep Suri, Hannu T. Huhdanpaa, Kathryn James, David S. Carrell, Curtis P. Langlotz, Nancy L. Organ, Eric N. Meier, Karen J. Sherman, David F. Kallmes, Patrick H. Luetmer, Brent Griffith, David R. Nerenz, Jeffrey G. Jarvik, Comparison of Natural Language Processing Rules-based and Machine-learning Systems to Identify Lumbar Spine Imaging Findings Related to Low Back Pain, Academic Radiology, Volume 25, Issue 11, 2018, Pages 1422-1432, ISSN 1076-6332, https://doi.org/10.1016/j.acra.2018.03.008.

Michael Jin, Marc Schröder, Victor E. Staartjes, 15 - Artificial Intelligence and Machine Learning in Spine Surgery, Editor(s): Anand Veeravagu, Michael Y. Wang, Robotic and Navigated Spine Surgery, Elsevier, 2023, Pages 213-229, ISBN 9780323711609, https://doi.org/10.1016/B978-0-323-71160-9.00015-0.

The above are meant as indicative of the bibliography which needs to be systematically categorized and analyzed. Potentially, X-ray, MRI and CT modalities could be considered so as to understand the SOA in the field.

In regards to the proposed research methodology, unfortunately, the use of handcrafted features (measurements) coupled with machine learning (ML) classifiers represents a move towards the past, as indeed, modern developments in the field of medical imaging deploy deep learning. Example architectures for deep learning are transformers and CNN-based architectures. The benefit of using this is that image-based features are automatically extracted and thus, there is no objectivity in the choice of measurement points. Moreover, to incorporate the diagnostic information within the performance of the deep learning system, the authors could use class activation maps, which will associate the presence of a specific condition to certain regions within the image. To conclude the existing approach is not SOA, which raises concerns with respect to its impact within the community.

Section 2.2 should be expand by adding references and explaining the complications of the conditions, and treatment. A figure demonstrating where these conditions occur and how they vary from each other would be helpful.

In Section 2.3, it is not clear how the 10 measurement points are identified. It appears that the procedure is not automated. How do variations from manual annotation are dealt with? What happens when there is low contrast and/or artifacts in the images, and thus, there is noise complicating their selection?  Moreover, it is evident from Table 1, that the statistical means of the measurements between the two classes are quite close to each other, which further reinforces the issue of the impact of noise/objectivity.

In Section 2.4, it is not clear how the machine learning methods were selected. What are the criteria? Support Vector Machines and K-NN would be more approriate classifiers to Logistic Regression (inc LASSO and Ridge regression). It is not clear why the authors considered this interesting problem in its binary form (presence of LDBH or not) rather than examining its multi-class alternative, whilst having a small number of training samples, which would make it scientifically challenging.

In Section 3, the explanation of feature/measurement importance is rather basic. Instead, a variety of modern approaches, such as recursive feature elimination/addition, permutation importance, univariate feature selection, and Boruta, should be explored to shed light on the impact of the measurements on the classification results.

Moreover, the authors should provide details of the means that the hyperparameters of the ML methods were tuned, and moreover, provide the implementation details (e.g., architecture, parameters, etc) so that the approaches can be reproduced.

A major weakness of the proposed approach is that there is no comparison with the state-of-the-art. This is justified due to the lacking of using X-ray for LDBH diagnosis. It is suggested that a means to perform comparison is to use an automated MRI-based image analysis approach (as the MRI studies are available) and perform comparison with the X-ray based approach.

Comments on the Quality of English Language

The language of the manuscript requires substantial improvement. There are numerous issues throughout the manuscript, which requires careful editing . A simple example from the abstract is as follows:

In this study, five machine  learning methods, including lasso regression, ridge regression, decision tree, random forest and eXtreme Gradient Boosting was application and integration to identify important variables in predicting LDBH from lumbar spine X-rays.

where the words in bold should become "were applied and integrated". It is also unclear what the term integrated means in the context of the sentence.

Author Response

(The authors gave the same response as above.)

Round 2

Reviewer 1 Report

Comments and Suggestions for Authors

Dear authors,

Thank you for submitting your work. I understand that you have put in a lot of effort to improve the study. However, I agree with you that studies with a low number of cases and significant heterogeneity regarding gender and age groups lack scientific validity. Such studies require an adequate number of cases to lead to clear scientific evidence aimed at advancing knowledge.

While I understand the difficulties in gathering data and the costs of research projects, these reasons cannot justify the acceptance of a scientific work that should be based on a solid scientific design and an adequate number of samples. I appreciate your interest in advancing knowledge and encourage you to continue your research with a larger sample size.

Author Response

(The authors gave the same response as above.)

Reviewer 2 Report

Comments and Suggestions for Authors

1 Authors should give more detailed description of Figure 3.

2 Please improve the quality of Figure 4.

3 When writing phrases like “Current medical practices incorporate ML methods, which play a crucial role by extracting valuable insights from data without the need for predefined human rules” it must cite related works (10.1155/2023/2345835; 10.1117/12.2540175) in order to sustain the statement.

4 Could you please check your references carefully? All references must be complete before the acceptance of a manuscript.

Author Response

(The authors gave the same response as above.)

Reviewer 3 Report

Comments and Suggestions for Authors

The authors made a substantial effort in accommodating my comments. From the point of view of using machine learning methods, the proposed methods and their use are standard, and do not contribute to the state of the art. However, there is value in making X-ray based diagnostics available for disc abnormalities, with the clinical significance and impact to patients being substantial. My recommendation to accept this manuscript lies indeed within the context of its clinical significance, which aligns well with the aim and objectives of the journal.

Author Response

(The authors gave the same response as above.)
